# Mycoextraction: Rapid Cadmium Removal by Macrofungi-Based Technology from Alkaline Soil

**Miaomiao Chen** [1,2,†]**, Likun Wang** [1,†]**, Junliang Hou** [3]**, Shushen Yang** [1,2]**, Xin Zheng** [1]**, Liang Chen** [1] **and Xiaofang Li** [1,*] 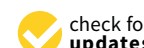

[1] Key Laboratory of Agricultural Water Resources, Centre for Agricultural Resources Research, Institute of Genetics and Developmental Biology, Chinese Academy of Sciences, Shijiazhuang 050021, China; miao7872209@163.com (M.C.); lkwang@sjziam.ac.cn (L.W.); ssyang@sjziam.ac.cn (S.Y.); zhengxin@sjziam.ac.cn (X.Z.); cl007z@aliyun.com (L.C.)

[2] University of Chinese Academy of Sciences, Beijing 100049, China

[3] Hebei Geoenvironment Monitoring, Shijiazhuang 050021, China; houjunliang021@163.com

\* Correspondence: xfli@sjziam.ac.cn; Tel.: +86-311-85822874

† These authors contributed equally to this work.

**Abstract:** Fungi are promising materials for soil metal bioextraction and thus biomining. Here, a macrofungi-based system was designed for rapid cadmium (Cd) removal from alkaline soil. The system realized directed and rapid fruiting body development for subsequent biomass harvest. The Cd removal efficiency of the system was tested through a pot culture experiment. It was found that aging of the added Cd occurred rapidly in the alkaline soil upon application. During mushroom growth, the soil solution remained considerably alkaline, though a significant reduction in soil pH was observed in both Cd treatments. Cd and dissolved organic carbon (DOC) in soil solution generally increased over time and a significant correlation between them was detected in both Cd treatments, suggesting that the mushroom-substratum system has an outstanding ability to mobilize Cd in an alkaline environment. Meanwhile, the growth of the mushrooms was not affected relative to the control. The estimated Cd removal efficiency of the system was up to 12.3% yearly thanks to the rapid growth of the mushroom and Cd enrichment in the removable substratum. Transcriptomic analysis showed that gene expression of the fruiting body presented considerable differences between the Cd treatments and control. Annotation of the differentially expressed genes (DEGs) indicated that cell wall sorption, intracellular binding, and vacuole storage may account for the cellular Cd accumulation. In conclusion, the macrofungi-based technology designed in this study has the potential to become a standalone biotechnology with practical value in soil heavy metal removal, and continuous optimization may make the system useful for biomining.

**Keywords:** cadmium; mycoextraction; oyster mushroom; fruiting body; substratum; soil heavy metal removal; transcriptomics

## 1. Introduction

Heavy metal pollution in soil and water has raised concerns due to the carcinogenic nature of heavy metals [1]. Cd is a rare element of high toxicity. It is commonly present in wastewater effluent from electroplating, battery manufacturing, fertilizers, and pesticides [2]. Cd is non-biodegradable and easily transfers to the food chain, posing considerable risks to human health. Remediation of soil and water Cd pollution is now a hot topic for environmental scientists worldwide.

Several fungal strains were found to possess the extraordinary ability of heavy metal tolerance. For instance, *Penicillium* spp. could survive in a yeast extract agar medium with up to 1500 mg/kg

nickel and 3000 mg/kg cobalt [3]. *Trichoderma harzianum* has a minimum inhibitory concentration (MIC) of 35 mg/L Cd in potato dextrose agar (PDA) medium [4]. Many more Cd-tolerant fungi have been reported [5–7]. Meanwhile, the dry biomass of *Ganoderma carnosum* [8] and *Pleurotus ostreatus* [9] can absorb lead from water with a maximum biosorption capacity of up to 22.79 mg/g and 121.21 mg/g, respectively. These fungal strains are thought to be promising biosorbents for heavy metal removal from wastewater.

Heavy-metal-tolerant fungi have also been considered for soil heavy metal remediation. Conventionally, they are used as augmented bioagents to facilitate phytoremediation. Mycorrhizal fungi were found to improve heavy metal accumulation in plant tissues and the phytoremediation performance of marigold [10] and maize [11]. In another study, inoculation of *Piriformospora* and *Glomus mosseae* significantly reduced the Cd content in wheat shoots [12]. A limited number of studies have indicated the possibility of using macrofungi for soil heavy metal removal. In a study to test the ability of *Clitocybe maxima*in to degrade trichlorophenol, Liu et al. [13] found that the fruiting body of the strain was able to absorb a considerable amount of heavy metals. Similar results were found in the remediation of Cd-phenanthrene co-contaminated soil by *Pleurotus cornucopiae* [14]. Damodaran et al. [6] first proposed the standalone use of macrofungi fruiting body to remove heavy metals from acidic soil, using a hyper-accumulator mushroom strain as a heavy metal extractor. However, their lack of design for fruiting body development led to a low growth rate and stringent requirements for a favorable environment for fruiting body formation, making their scheme impractical for soil heavy metal removal.

Currently, the exploration of heavy-metal-tolerant fungal strains with desired traits is still an important task. More importantly, engineering a design for taking advantage of the rapid growth and broad adaptability of macrofungi is required to make the fungal-based technology practical for soil heavy metal removal. We previously identified a Cd-tolerant oyster mushroom strain with moderate capacity of Cd accumulation in its mycelia but good adaptability for fruiting body formation [15]. In this study, we presented a novel design for a mushroom bagging strategy, realizing directed fruiting body formation and rapid growth. Its Cd removal efficiency was tested in a pot culture experiment. Mechanisms related to the Cd sorption process were explored by monitoring soil solution chemistry and transcriptomic analysis of the fruiting body. In all, the macrofungi-based technology presented here has potential to become a standalone biotechnology with practical values for soil heavy metal removal, and continuous optimization through strain improvement and soil conditioners may make this system useful for biomining.

## 2. Methods and Materials

### 2.1. Strains and Soil

An oyster mushroom strain, JINONG 21, was purchased from the local market and used as a metal extractor in this study. Its phylogenetic affiliation was identified to be *Pleurotus ostreatus* using an internal transcribed spacer as a molecular marker [15]. This strain was found to be highly Cd-tolerant. Its mycelia have a minimal inhibitory concentration of 4 mg Cd/L in PDA medium, and can accumulate up to 970 mg Cd/kg dry biomass. According to the user manual, the oyster strain requires low atmospheric temperature under 22 °C and scattered light for induction of fruiting body formation.

The soil used in this study was collected from the campus area of the Centre for Agricultural Resources Research, Chinese Academy of Sciences in Shijiazhuang, China. The soil was air-dried and finely sieved through a 2-mm mesh. Soil physicochemical properties were then measured. Briefly, the soil organic matter content (OMC) and cation exchange capacity (CEC) were analyzed following the method of Li et al. [16]. Electrical conductivity (EC) and pH were determined in water extracts (1:5 *v/v*) electronically. Total nitrogen (N) content was determined as $N_2$ by thermal conductivity

detection [17]. Soil-available phosphorus (P) content, available potassium (K) and total Cd content were determined by inductively coupled plasma mass spectrometry (ICP-MS) [18].

### 2.2. Cd Aging in the Soil

Aging effect on Cd mobility in the soil was tested, by monitoring Cd content dynamics in soil solution. Briefly, 300 g dry soil was placed into a beaker and artificially contaminated with Cd in the form of $CdCl_2 \cdot 2.5H_2O$. Additional water was added to adjust the soil moisture to 80% maximum water holding capacity. Two Cd treatments, 10 mg/kg and 150 mg/kg, which were also applied for subsequent mycoextraction tests, were examined together with the control. Soil solution sampler (Rhizosphere Research Products, Wageningen, The Netherlands) was used for soil solution sampling once a day since the third day post-incubation, until the soil solution Cd tended to be stable. All tests were done at 25 °C.

### 2.3. Removable Mushroom Bagging Design and Conditions for Fruiting Body Development

An efficient metal extractor for practical use must possess the ability of rapid growth and be easily harvested. Different bagging strategies (with vs. without bagging) and plantation mode (semi-underground vs. underground) were tested for their impacts on the rate of fruiting body development. Cottonseed shells (78%, *w/w*) and wheat bran (20%) were mixed with sucrose (1%) and gypsum powder (1%) and used as substrates for mushroom cultivation. Water was added to adjust the moisture of the bagged substrates to 60%. The substrate bags were then sterilized and inoculated with oyster mushroom mycelia, followed by 25 days incubation in the dark at 28 °C to allow the complete overgrowth of mycelia in the substrate bags.

Complete removal of the plastic bag, coupled with underground planting of the substrate with overgrown mycelia (i.e., mushroom bags), failed to develop a fruiting body within a month under the standard conditions. A special design for the bagging scheme was then developed, as shown in Figure 1. Two-thirds of the plastic bag was replaced by a plastic mesh bag, which was planted into the soil, leaving one-third of the mushroom bag aboveground. A small hole located on the top of the bag allowed the directed growth of the fruiting body. This bagging design made fruiting body formation much easier, and the directed growth was rapid and easy. Moreover, this design makes the substratum part removable, which was found to be an important Cd reservoir, as detailed in the Sections 3 and 4.

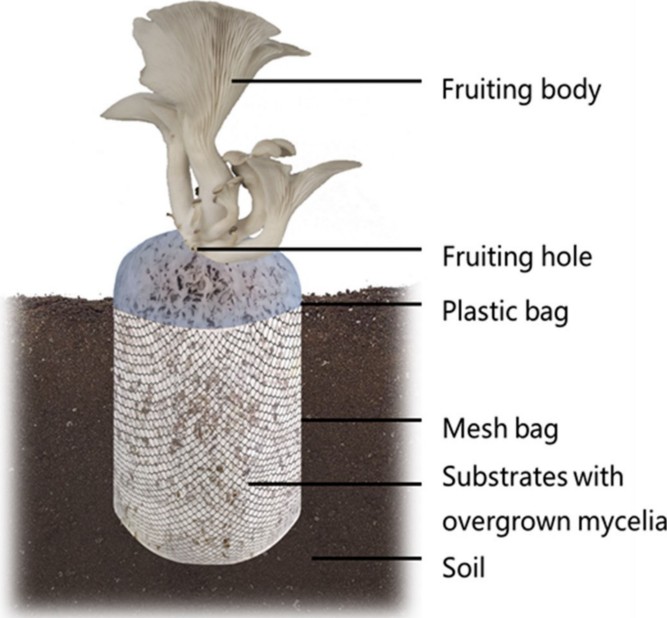

**Figure 1.** A schematic diagram of the mycoextraction system designed in this study.

### 2.4. Pot Experiment of Cd Mycoextraction

Cadmium removal efficiency of the mycoextraction system was tested in Cd-contaminated soil under greenhouse conditions. Briefly, a 0.35 kg mushroom bag that was overgrown with mycelia was planted semi-underground into the Cd-contaminated soil. For each pot, 1 kg soil was used and incubated at $18 \pm 1\,^\circ\text{C}$ for up to 35 days. During cultivation, deionized water was added to the pots to compensate for the water loss, maintaining approximately 65% of the maximum water holding capacity. All pots were randomly placed in the greenhouse.

Three batches of fruiting body were harvested on the 7th, 21st and 35th day since incubation. Aliquots of fresh mushroom fruiting body were sampled from all pots at the end, then stored in $-80\,^\circ\text{C}$ refrigerator and subjected to RNA extraction later. For chemical analyses, the fruiting body biomass and substratum residue were washed with deionized water and oven-dried at $60\,^\circ\text{C}$.

Soil solution of each pot was sampled every seven days since incubation started, using a soil solution sampler (Rhizosphere Research Products, Wageningen, The Netherlands) as described above. The soil after mushroom bag harvest was also sampled for pH determination.

### 2.5. Chemical Analysis

Cadmium in soil solution was determined by using ICP-MS (NexION 300Q, PerkinElmer, Waltham, MA, USA). Briefly, dried soil was ground and sieved through 100 mesh sieve. Around 0.10 g soil was digested with 10 mL of *aqua regia* (2.5 mL $HNO_3$ + 7.5 mL HCl) in an Anton Paar microwave digestion system (Multiwave PRO, Anton Paar, Graz, Austria). The digested sample was transferred into a 10-mL volumetric flask and diluted to 10 mL with 3% nitric acid solution for Cd determination [18]. DOC was determined by a TOC Analyzer (Vario TOC select, Elementar, Langenselbold, Germany). Cd in fruiting body and substratum was determined using a Zeenit 700 P Atomic Absorption Spectrometer (Analytik Jena, Jena, Germany) equipped with a flame atomizer after acid digestion.

### 2.6. Transcriptomic Sequencing

Frozen fruiting body tissue (ca. 200 mg) was ground to a fine powder in liquid nitrogen and total RNA was extracted using TRIzol reagent (Invitrogen, Carlsbad, CA, USA) following the manufacturer's instructions. RNA was immediately purified by using RNeasy MinElute Cleanup Kit (Qiagen, Germantown, MD, USA). The quantity of the isolated RNA was examined using a Nanodrop ND-2000 (Thermo Scientific, Waltham, MA, USA) and verified through agarose gel electrophoresis. The integrity of total RNA was measured through Agilent2100 (Agilent Technologies Santa Clara, CA, USA). RNA Integrity Numbers (RIQNs) from 1 to 10 were assigned to each sample to indicate its integrity and quality. All samples' RIN were above 9.9, which qualified for cDNA library preparation. One milligram of total RNA was enriched for mRNA using oligo-dT attached Magnetic beads (Invitrogen, Waltham, MA, USA) with oligo (dT). The cDNA library was constructed using the extracted mRNA with the Truseq TM RNA sample prep kit (Illumina, San Diego, CA, USA). Sequencing was conducted using the Illumina HiSeq 4000 platform provided by Majorbio Bio-Pharm Technology Co., Ltd., Shanghai, China.

### 2.7. Reference Mapping and RNA-Seq Analysis

Nine sequence data assemblages containing 458.2 million raw reads in total were generated by the Illumina Hi-Seq 4000 instrument (Illumina, San Diego, CA, USA). Adapters were trimmed off by using SeqPrep (omicX, Le-Petit-Quevilly, France) and raw data was subsequently passed through quality trimming by using Sickle (omicX, Le-Petit-Quevilly, France). Reads with incorrect called bases towards 3′ end and 5′ end were trimmed off. Phred score (Q20 and Q30), GC-content, and sequence duplication level of the clean data were calculated. The high-quality clean data obtained were used in reference mapping.

The *Pleurotus ostreatus* reference genome [19] was downloaded from National Center for Biotechnology Information (NCBI). The genome sequence file consists of 13 contigs. To ensure the accuracy of our analysis, no more than five mismatches were allowed in the alignment. The alignment data were utilized to calculate the distribution of reads on the reference genes and to perform the coverage analysis. Outputs of the sequence alignment containing the aligned reads and mapping information were used for the downstream analyses. Data analysis was performed on the online platform of Majorbio I-Sanger Cloud Platform (Majorbio, Shanghai, China).

### 2.8. Differentially Gene Expression Analysis

In order to identify the transcriptomic Cd response of the strain used in this study, differentially gene expression analysis was performed. Gene expression levels were estimated by using RSEM [20] for each sample. Clean data was mapped to the reference genome, and read counts for each gene were obtained from the mapping results. Read counts for each sample were normalized using the number of fragments per kb of exon region per million mappable reads (FPKM). Differentially expressed genes (DEGs) were obtained by comparing each two Cd concentrations (0 vs. 10, 10 vs. 150, 0 vs. 150) through DESeq2 [21]. A minimum of two-fold expressional difference (i.e., log2 (fold change) $\leq -1.0$ or $\geq 1.0$) in the paired libraries and a corrected *p* value (*p* adjust) of 0.05 or less were used as standards to judge each identified gene. The identified DEGs were saved for function annotation.

### 2.9. Functional Annotation of Candidate Genes

All DEGs were classified and annotated using Gene ontology (GO) term enrichment analysis [22] and Kyoto Encyclopedia of Genes and Genomes (KEGG) pathway enrichment analysis [23] at the threshold of corrected $p < 0.05$. The protein functional category for DEGs was assigned using the NCBI KOG (Eukaryotic Orthologous Group) database [24]

### 2.10. Statistical Analysis

All comparisons were subject to one-way analysis of variance (ANOVA) using SPSS 16.0 software (version 16.0, IBM, New York, NY, USA). Means separation was conducted by using Duncon's multiple range test, with $p \leq 0.05$ considered significant.

## 3. Results

### 3.1. The Soil

The soil used in this study was a typical fluvo-aquic soil widely distributed in north China. It has an extremely alkaline pH of 8.9, total N of 0.035%, available P of 0.0014%, organic matter content of 4.3 g/kg, and total Cd of <0.01 mg/kg. The maximum water holding capacity of the soil was determined to be 45 mL water per 100 g soil.

Aging effect was monitored up to 13 days at eight sampling time points. In the 10 mg/kg treatment, soil solution Cd decreased from 27.8 mg/L (theoretically calculated based on the total Cd and water added; the same below) at the beginning to 0.54 μg/L at the 3rd day, with a continuous decrease to 0.21 μg/L at the 13th day. In the 150 mg/kg treatment, soil solution Cd decreased from 416.7 mg/L at the beginning to 24.87 μg/L at the 3rd day, with a continuous decrease to 12.6 μg/L at the 13th day (Figure 2). No significant difference was found in soil solution Cd among all the sampling timepoints of the 10 mg/kg treatment, while a significant difference was found between the 3rd and 6th day in the 150 mg/kg treatment. Soil solution Cd tended to be stable after the 6th day in the 150 mg/kg treatment.

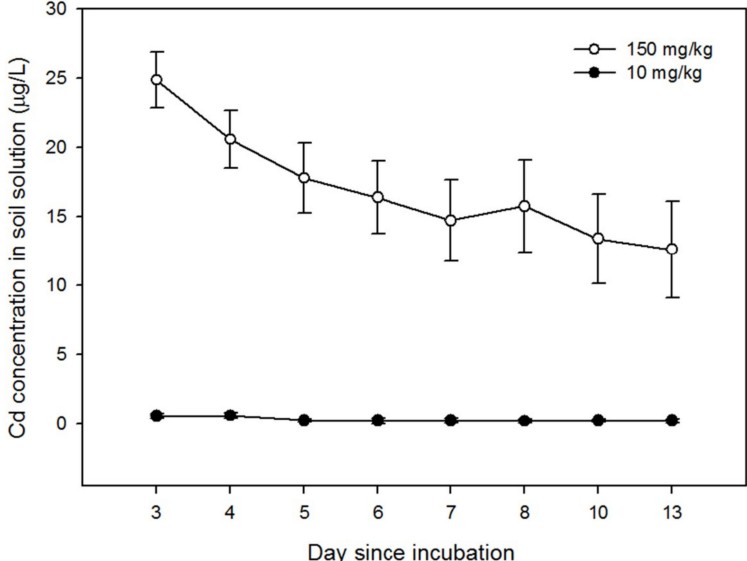

**Figure 2.** Cd aging curve in the alkaline soil used in this study. Soil solution was sampled (*n* = 3) once a day since the third day post-incubation until the soil solution Cd tended to be stable.

### 3.2. Dynamics in Mushroom Biomass and Biomass Cd

Three batches of fruiting body biomass were sampled during the 35 days of cultivation. Around 14 g dry biomass in total were obtained per pot, and there was no significant difference detected among all treatments (*p* > 0.05) (Figure S1).

For both Cd treatments, fruiting body Cd content increased substantially over time. At the 1st sampling time point, fruiting body Cd was low in all treatments and non-detectable in the control and 10 mg/kg treatment. For the 2nd and 3rd batches, fruiting body Cd increased substantially in the two Cd treatments with up to 4.3 mg Cd/kg dry biomass in the 150 mg/kg treatment, though no significant difference in Cd content was detected between the 2nd and 3rd batches of fruiting body for both the Cd treatments. Specifically, a 48-fold increase in fruiting body Cd was detected in the 2nd sampling in the 150 mg/kg treatment relative to the 1st sampling (Figure 3A). Substrates were sampled as well for Cd determination at the end of the cultivation. The amount of Cd accumulated by the substratum was up to 4.88 mg Cd/kg dry biomass in the 150 mg/kg treatment (Figure S2).

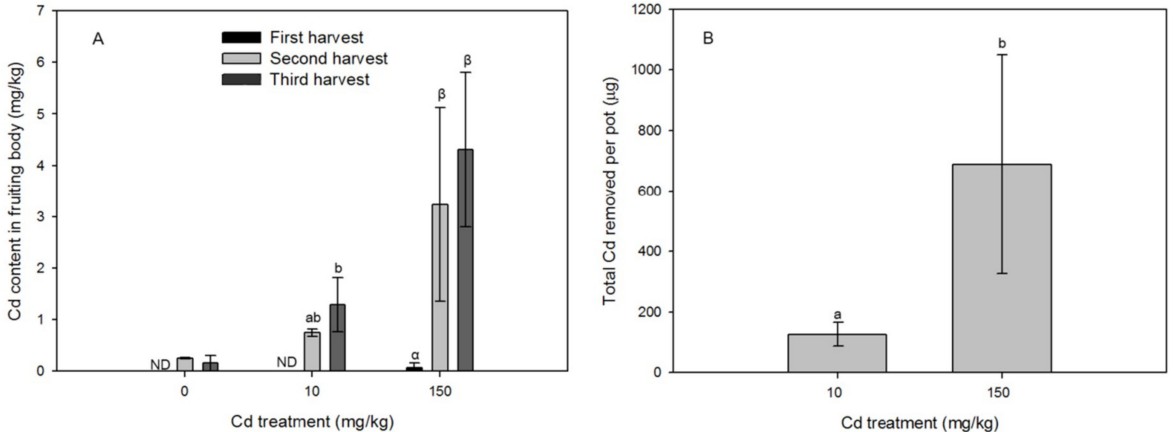

**Figure 3.** Dynamics in fruiting body Cd content in the two Cd treatments and the control over the 35-day cultivation (**A**), and the total Cd removal per pot calculated based on the Cd content of fruiting body and substratum (**B**). Bars with different letters are significantly different at *p* ≤ 0.05 (One-Way ANNOVA). ND, not detected.

Total Cd removal by the whole macrofungi system (fruiting body and substratum) was calculated. On average, each bagging system was able to remove 126 ± 38 µg and 688 ± 361 µg Cd in the 10 mg/kg and 150 mg/kg treatments, respectively, and a significant difference was detected between the 150 mg/kg treatment and 10 mg/kg treatment (Figure 3B).

### 3.3. Dynamics in Soil Chemistry

Soil pH before and after mushroom cultivation was measured (Figure 4). DOC and Cd contents in soil solution were monitored every seven days. The results showed that mushroom growth significantly reduced soil pH in all treatments including the control, yet the soil pH remained above 8.5 and the reduction was only around 0.4 in all treatments.

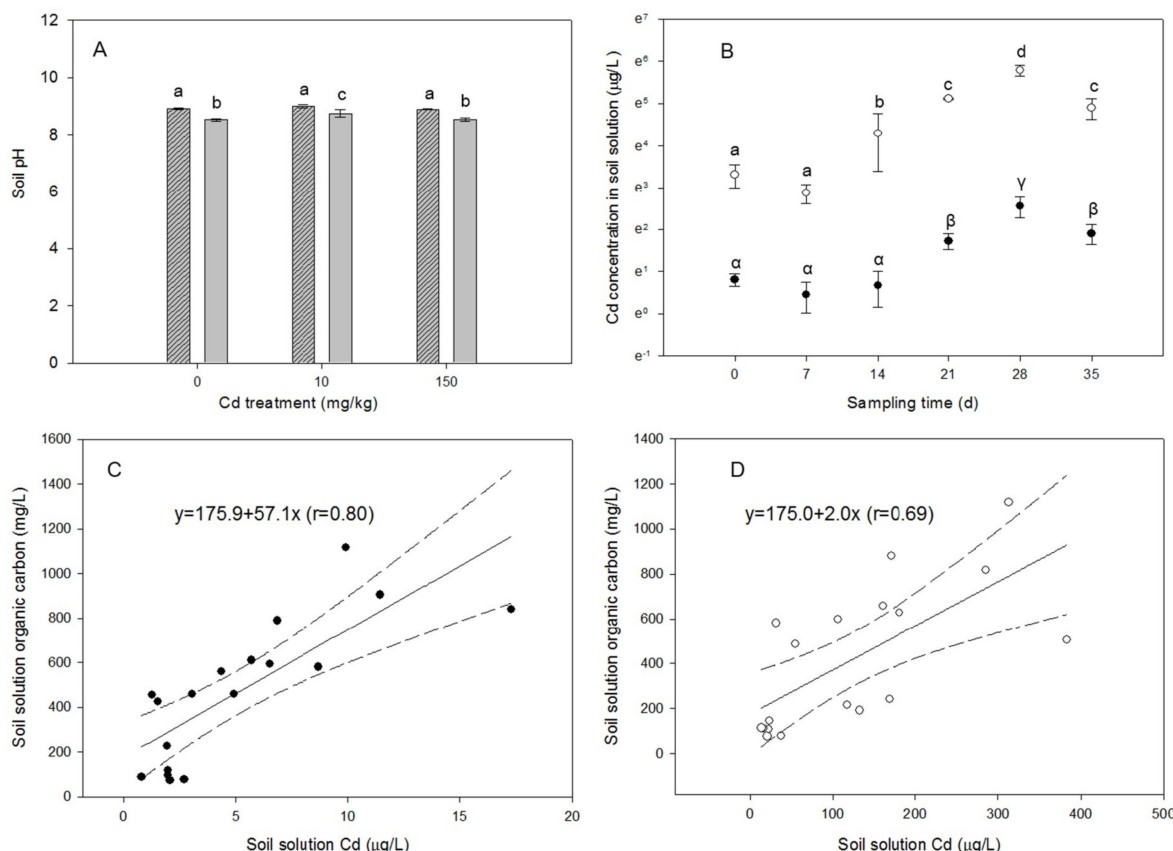

**Figure 4.** Dynamics in soil solution chemistry of the pot culture system in this study. Notes: Figure 4A, grid column stands for pH before planting, and grey column stands for pH after harvest; Figure 4B–D, cycles stand for the 150 mg/kg treatment, and dots stand for the 10 mg/kg treatment.

Cd concentration in soil solution generally increased over time in the two Cd treatments in parallel. Peak Cd was found to be 12.90 ± 3.16 µg/L and 327.10 ± 41.19 µg/L at the 28th day in the 10 mg/kg and 150 mg/kg treatments, respectively. Soil solution Cd tended to decrease at the 35th day in both Cd treatments. Soil solution Cd remained undetectable in the control throughout the incubation period (Figure 4).

A similar pattern was detected for soil solution DOC dynamics in both Cd treatments. Peak DOC content in soil solution was detected at the 28th day to be 906.15 ± 84.13 mg/L, 954.43 ± 118.24 mg/L, and 814.04 ± 249.80 mg/L in the 0 mg/kg, 10 mg/kg and 150 mg/kg treatments, respectively. No significant difference was found among the peak soil solution DOC ($p > 0.05$).

Taking all time points together, soil solution Cd and DOC were correlated significantly in both the Cd treatments (Figure 4).

### 3.4. Data Quality of the Transcriptomic Sequencing

RNA sequencing in this study produced 43.9–58.5 million paired-end reads in all samples after quality control. Q20, Q30, GC content, and error rates of the clean data were calculated (Table S1). A Phred score of Q30 above 94.78% was obtained, which indicated that the sequencing quality was satisfactory and can be used for subsequent quantitative analysis of expression levels.

### 3.5. Mapping Statistics and DEG Identification

Overall, 70.64–74.32% of the reads were mapped to the reference *P. ostreatus* genome. In the total mapped reads, 96.6–98.18% reads were uniquely mapped, which includes 91.98–92.5% mapped to coding sequence (CDS), 1.78–1.98% mapped to intron, and 0.21–0.24% mapped to intergenic regions (Table 1).

**Table 1.** Statistics of RNA-seq read mapping.

|  |  |  | No. of Reads | Percentage (%) |
|---|---|---|---|---|
| Total mapped |  |  | 31,654,173–42,895,795 | 70.64–74.32 |
|  | Multiple mapped |  | 674,534–1,210,904 | 1.82–3.40 |
|  | Uniquely mapped |  | 30,909,273–40,887,377 | 96.6–98.18 |
|  |  | CDS |  | 91.98–92.59 |
|  |  | Introns |  | 1.78–1.98 |
|  |  | Intergenic |  | 0.21–0.24 |
|  |  | 5′UTR |  | 2.45–2.92 |
|  |  | 3′UTR |  | 2.72–2.97 |
| Unmapped |  |  | 11,506,812–15,988,114 | 25.68–29.36 |
| Total reads |  |  | 43,930,516–58,542,646 |  |

DEGs were identified by comparing treatments 0 vs. 10, 10 vs. 150, and 0 vs. 150 mg/kg. In total, 1107, 2370, and 522 DEGs were identified for 0 vs. 10, 10 vs. 150, and 0 vs. 150 mg/kg, respectively. More downregulated DEGs were found relative to the upregulated DEGs when comparing the two Cd treatments against the control; conversely, more upregulated DEGs were found relative to the downregulated DEGs when comparing the high Cd treatment to the low Cd treatment. 40 identified DEGs were found in all the three comparing groups (Figure S3).

### 3.6. Function Enrichment Analysis

The top 20 upregulated DGEs and 20 downregulated DEGs in the two Cd treatments relative to the control were sorted out for further analyses. In the 150 mg/kg treatment, two upregulated DEGs (PLEOSDRAFT_1023504 and PLEOSDRAFT_1099120) were annotated as involved in metal efflux, two (PLEOSDRAFT_1056091 and PLEOSDRAFT_162256) for reactive oxygen species (ROS) responses, two (PLEOSDRAFT_1030898 and PLEOSDRAFT_1119533) for cell wall remodeling, and one (Hydph16) as hydrophobin, while downregulated DEGs were annotated to be involved in ROS induction (PLEOSDRAFT_152955 and PLEOSDRAFT_1092804), cell cycle (PLEOSDRAFT_1059138), cell wall remodeling (PLEOSDRAFT_1119533), peroxisomes proliferate (PLEOSDRAFT_1047466), DNA repair (PLEOSDRAFT_1101314), tricarboxylic acid cycle (TCA) (PLEOSDRAFT_1046185), and glyoxylate cycle (PLEOSDRAFT_29565).

In the 10 mg/kg treatment, three DEGs (PLEOSDRAFT_1113522, PLEOSDRAFT_1023504, and CorA3) were found to be involved in metal efflux, one DEG (PLEOSDRAFT_108474) involved in ROS response, two annotated as membrane transporters related to osmoregulation (PLEOSDRAFT_1077179) and phosphate transporting (PLEOSDRAFT_1042152), respectively. The downregulated DEGs included two DEGs (PLEOSDRAFT_1061253 and PLEOSDRAFT_34588) for aldo keto reductase, two (PLEOSDRAFT_1063163 and PLEOSDRAFT_1114418) for cell cycle,

one (PLEOSDRAFT_1034502) for cell wall remolding, one (PLEOSDRAFT_1113759) for heme-binding, and one (PLEOSDRAFT_1096954) for mitogen-activated protein kinase (MAPK) (Figure 5, Table S2).

Clusters of orthologous groups (COG) enrichment analysis showed that carbohydrate transport and metabolism, energy production and conversion and amino acid transport were significantly enriched in the 10 mg/kg treatment compared with the control (Table S3). Whereas, carbohydrate transport and metabolism, lipid transport and metabolism, and replication recombination and repair were significantly enriched in the 150 mg/kg treatment relative to the control (Table S3).

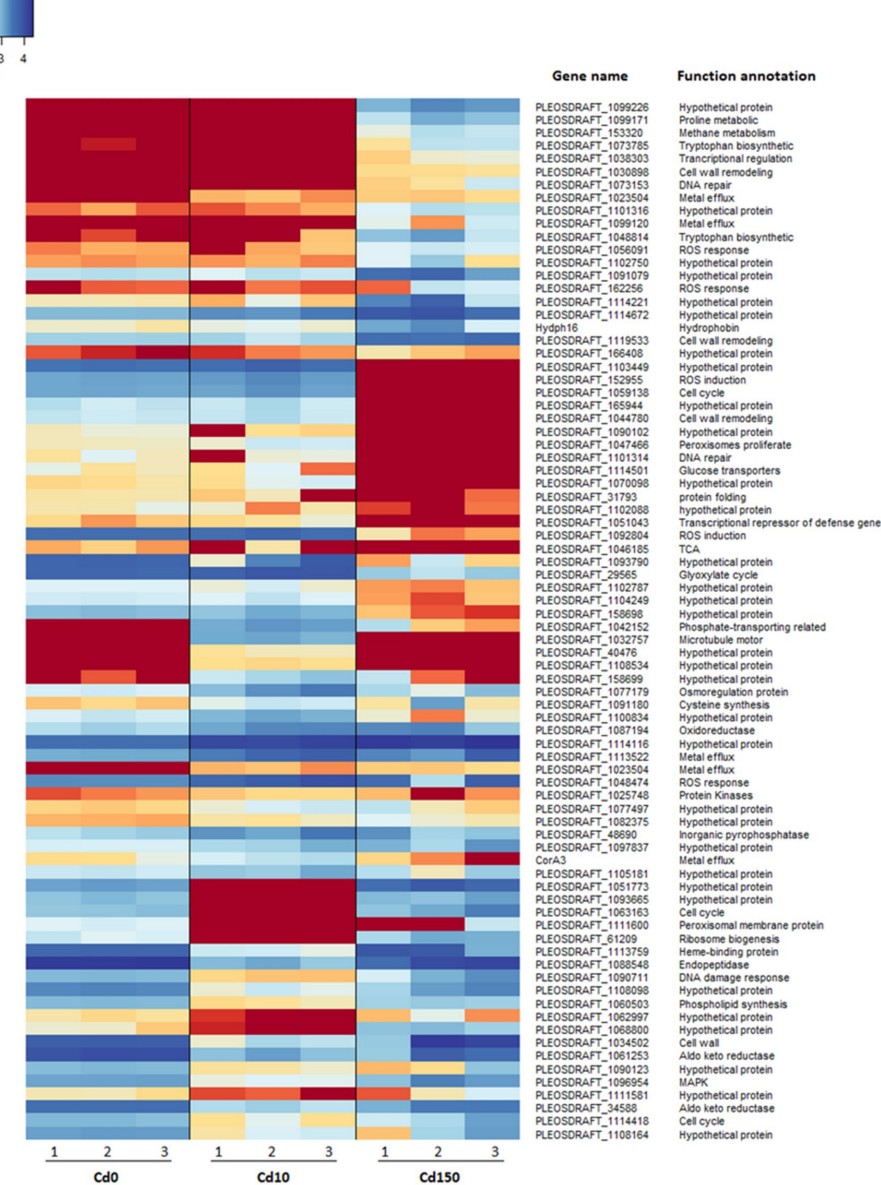

**Figure 5.** A heatmap showing the typical differentially expressed genes (DEGs) among the three treatments. The gene expression level increased with color from red to blue. Cd0, the control; Cd10, the 10 mg Cd/kg soil treatment; Cd150, the 150 mg Cd/kg soil treatment; NRF, normalized relative frequency. The numbers "1–3" represent the three biological duplicates. Gene name is designated with reference of the genome of *Pleurotus ostreatus* PC15 (PRJNA81933).

## 4. Discussion

A novel Cd mycoextraction system was designed, and its Cd removal efficiency was tested in an alkaline soil. Overall, the results showed that although the absolute Cd concentrations in both the

fruiting body and substratum were much lower than reported plant hyperaccumulators, the overall Cd removal rate by the mycoextraction system was considerably promising, thanks to the rapid growth and excellent metal mobilizing ability of the fungal strain in the alkaline soil. Considering that strain improvement for the oyster mushroom strain is much easier than plant breeding and much work can be done to modulate the soil pH, the mycoextraction system has potential to be a heavy metal removal approach of practical values and even a biomining tool.

### 4.1. Ability of the Macrofungi System in Cd Sorption in the Alkaline Environment

Phytoextraction of heavy metals from alkaline soil remains challenging due to the low metal bioavailability, despite the application of various metal mobilizers [25]. Likewise, the extremely alkaline soil pH may be a key factor limiting the Cd accumulation in the oyster mushroom fruiting body, as it was found in the current study. Generally, soil pH is a dominant parameter controlling heavy metal bioavailability [16,26]. Cd mobility decreased rapidly with increase pH. As reported by van Gestel and Koolhaas [27], increase in pH from 3.5 to 6.5 led to a 2–10 folds decrease in soil water-extractable Cd. Soil soluble Cd decreased rapidly when pH > 6.5 [28], which means that soil with alkaline pH presents a significant challenge to heavy metal hyperaccumulators. The aging experiment in this study also indicated that the added Cd was fixed rapidly by the soil and continued to attenuate until the 6th day since incubation. Nonetheless, such alkaline soils are actually widely distributed in China, including Hebei, Shanxi, Beijing, Inner Mongolia, Shandong and Xinjiang provinces [29], as well as many other countries. Bioremediation of such metal-contaminated soils has therefore long been considered as a major difficulty by soil restorationists.

The results in the current study showed that the macrofungi system may have an outstanding ability to mobilize Cd through pH reduction and DOC complexation (Figure 4), though we still cannot quantify the contribution of mushroom fruiting body to substrate degradation and soil pH reduction. Organic acid is a strong agent for metal complexation leading to metal desorption in neutral/alkaline soil [30]. External DOC was able to increase desorption of Cd, Zn and Ni by 2–6 folds in aquifer material [31]. Field studies have also documented that DOC is a key agent for Cd mobilization in various soils [32]. Organic carbon is also an important component of hyperaccumulator root exudates, which plays an important role in plant metal acquisition [33]. It is thus reasonable to conclude that the DOC released by the mushroom bag may have substantial contribution to Cd mobilization in the soil and Cd uptake by the fruiting body, since a parallel increase over time in DOC and Cd in soil solution was detected in this study (Figure 4).

The oyster mushroom strain used in this study is a typical Cd hyperaccumulator, which was able to accumulate 151.24–676.07 mg/kg Cd (dry biomass based; the same below) within 7 days in its mycelia in solid medium containing 4–20 mg/L Cd [15]. Unexpectedly, the actual biomass Cd in the fruiting body cultivated in the soil was found much lower than typical plant hyperaccumulator (Table 2). For example, *Sedum alfredii* was reported to accumulate above 176 mg/kg Cd in its shoot during 118 days growth [34]. However, literature exploration indicated that Cd accumulation capacity of the oyster mushroom in this study was comparable to the previous reported mushroom strains, such as *Oudemansiella radicata*, who accumulated 2.6–18.5 mg/kg Cd in its fruiting body in 63 days growth [35]. It is thus possible that macrofungal Cd hyperaccumulators determined based on mycelium Cd enrichment may not indicate a hyperaccumulator nature of their fruiting body. We speculate that the low Cd enrichment by the fruiting body in this study may be related to the cultivation mode of the mushroom bag. Mushroom bags may limit the contact of mushroom mycelia with soil Cd, since a study by Damodaran et al. [6] showed that *Galerina vittiformis* can accumulate 852 mg/kg Cd in its fruiting body when the strain was cultivated directly to the soil contaminated with 50 mg/kg Cd. It demonstrated that free-standing growth of mushroom in soil led to Cd accumulation in the fruiting body comparable to or even higher than plant hyperaccumulators.

**Table 2.** Cadmium adsorption capacity of selective plants and mushroom strains reported in recent years.

| | Species | Cd Concentration (mg/kg) | pH | Growth Time (day) | Accumulation Concentration (mg/kg Dry Biomass) | Cd Accumulation (µg) | Removal Efficiency per Year (%) | Reference |
|---|---|---|---|---|---|---|---|---|
| Plant | *Cyphomandra betacea* | 10 | 7.0 | 40 | Stem 23.46; Leave 24.95; Shoot 24.57 | Stem 12; Leave 39; Shoot 52 | 3.2 | [36] |
| | *Festuca arundinacea* | 500 | Unknown | 90 | Shoot 50 | Shoot 210 | 0.2 | [37] |
| | *Amaranthus hypochondriacus* | 5 | 7.2 | 60 | Shoot 146.5 | Unknown | 9.3 | [38] |
| | *Sedum alfredii* | 31.7 | 7.7–7.8 | 118 | Shoot 178.7–212.6 | Shoot 1676.5–2635.2 | 3.3–5.2 | [30] |
| | *Brassica napus* | 29.4–43.7 | 5.7 | 60 | Shoot 680.6–1221.0 | Unknown | 4.9–7.3 | [39] |
| | *Thlaspi caerulescens* | 2.9 | 7.3 | 60 | 47.4 | 35 | 3.7 | [40] |
| Macrofungi | *Oudemansiella radicata* | 5–30 | 7.5 | 63 | 2.6–18.50 | 69.8–437.9 | 1.6–1.9 | [31] |
| | *Tricholoma lobynsis* | 12 | 7.5 | 45 | 13.67 | 31.7 | 0.3 | [41] |
| | *Pleurotus cornucopiae* | 5–20 | 7.42 | Unknown | 0.7–3.8 | 11.6–59.9 | Unknown | [14] |
| | *Coprinus comatus* | 50 | 7.5 | Unknown | 2.8 | 28.2 | Unknown | [42] |
| | *Clitocybe maxima* | 30 | 6.33 | >35 | 2.5 | 60 | >0.5 | [13] |
| | *Armillaria mellea* | 0.24, wild | Unknown | Unknown | 1.14 | Unknown | Unknown | [43] |
| | *Polvporus squamosus* | 0.56 | Unknown | Unknown | 32.4 | Unknown | Unknown | |
| | Oyster mushroom | 150 | 8.92 | 35 | Fruiting body 4.3; substrate 4.88 | Fruiting body 56.3; substrate 660 | 4.6 | This study |
| | | 10 | 8.92 | 35 | Fruiting body 1.3; substrate 0.87 | Fruiting body 8.96; substrate 118 | 12.3 | |

Notes: Removal efficiency = (Cd total amount removed in biomass/Cd total amount in soil) × 100%.

### 4.2. Cd Removal Efficiency of the Macrofungi System

An ideal bio-extraction tool for soil heavy metal removal should be easily cultivated, harvestable, and of high biomass and favorable metal enrichment capacity. Plant materials for phytoextraction are developed decades ago and are currently the only available bio-extraction scheme [44,45]. Mushrooms feature in their rapid growth and extreme and broad heavy metal resistance, and have been proposed as a heavy metal bio-extraction tool years ago [6,46]. Both high Cd accumulation and fruiting body formation were realized in these studies, yet lack of engineering design for rapid growth and fruiting body development, a key for bio-extraction efficiency, limited the practical values of their systems as mycoextraction. In this study, we used the mushroom bag and semi-belowground plantation as a cultivation mode. Meanwhile, a special bagging method was designed for both belowground nutrient and metal uptake and directed fruiting body development. These engineering designs realized rapid growth and made the fruiting body easily harvested. In addition, the strain used here is a most widely cultivated mushroom worldwide, with good adaptability and easy fruiting body formation.

Unexpectedly, Cd accumulation in the mushroom bag contributed to Cd removal substantially, with both a higher Cd concentration and a higher total Cd accumulation than the fruiting body part (Table 2). It is however not able to determine whether Cd accumulated in the mushroom bag substrate were mainly fixed by organic matter sorption or by the absorption of oyster mushroom mycelia. Litter without grinding and modification is *per se* not good biosorbents for heavy metals, while environmental engineering measures may make the mushroom bag important for higher heavy metal sorption. For example, use of modified agricultural residues may largely improve the Cd accumulation of the system, since it has been reported that modified paddy straw was able to absorb around 1.5% Cd ($w/w$) [47], and fine power of *Salvinia* biomass can absorb up to 4% Cd [48], tested in solution under laboratory conditions. It is also worth noting that bioavailability is a factor of not only the total metal concentration in the environmental medium, but also the mass transfer process. Performance of biosorbents on metal sorption may be quite different in soil from in water where they were tested under ideal conditions. Direct evidence for the control of mass transfer process on soil metal bioavailability is that increase in soil moisture directly led to an increase in metal extractability, mainly through the carriage effect of DOC and labile clays [32,49]. Therefore, for any bio-extraction system, engineering efforts are required to not only elevate the tolerance and sorption capacity of the bio-materials, but also the metal mobility and mass transfer for metal bioavailability.

Cd accumulation by both the fruiting body and the substratum was not impressive, while the overall Cd removal rate by the system was calculated to be outstanding relative to the available phytoextraction and mycoextraction studies (Table 2), thanks to the rapid growth of the mushroom and substantial contribution of Cd sorption by the mushroom bag. The mycoextraction method developed here is thus of practical values for rapid soil heavy metal removal, and continuous optimization of the system may also make the tool useful in biomining if heavy metal accumulation by the removable part can be improved remarkably.

### 4.3. Transcriptomic Response of the Oyster Mushroom to Cd

Transcriptional sequencing provided insight into the cellular response of the oyster mushroom at the fruiting body development stage, under both the low and high Cd stress. Here we discussed mainly the transcriptional responses under high Cd stress, though it shared a large portion of induced cellular processes between the two Cd treatments. The results showed that various pathways were induced under Cd stress, prominent ones including metal transport, cell wall remodeling, cysteine-rich related genes binding and ROS response (Figure 6, Table S4). Some specific transporters may actively be involved in the oyster mushroom Cd detoxification/accumulation. The most upregulated metal transporters were affiliated with the Major Facilitator Superfamily (MFS), including a typical ABC transporter. MFS is the largest transporter group, which might mediate the transport of relevant secondary metabolites across membranes via facilitated diffusion, symport, or antiport [50]. It is difficult to predict whether an MFS is acting as an importer or an exporter [51]. Some MFS genes could

increase Zn and Cd tolerance, through metal sequestration to vacuoles [52] or efflux [51]. Transporters for $Ca^{2+}/H^+$, $K^+$ and $Na^+$ were also found to be upregulated significantly. CAX, a $Ca2^+/H^+$ exchanger, plays a role in maintaining a low Cd concentration in cytosolic usually through sequestration into the vacuole [53]. The potential roles of $K^+$ and $Na^+$ transporters in Cd stress response remains unknown to our knowledge. Multidrug and toxic compound extrusion (MATE) and inorganic phosphate transporters were also implied to be involved in Cd stress response. MATE has been reported to regulate Cd detoxification via exporting Cd to outside of cytoplasm in plant [54,55]. Phosphate transporters are well known as Cd-complex agents for Cd export [56].

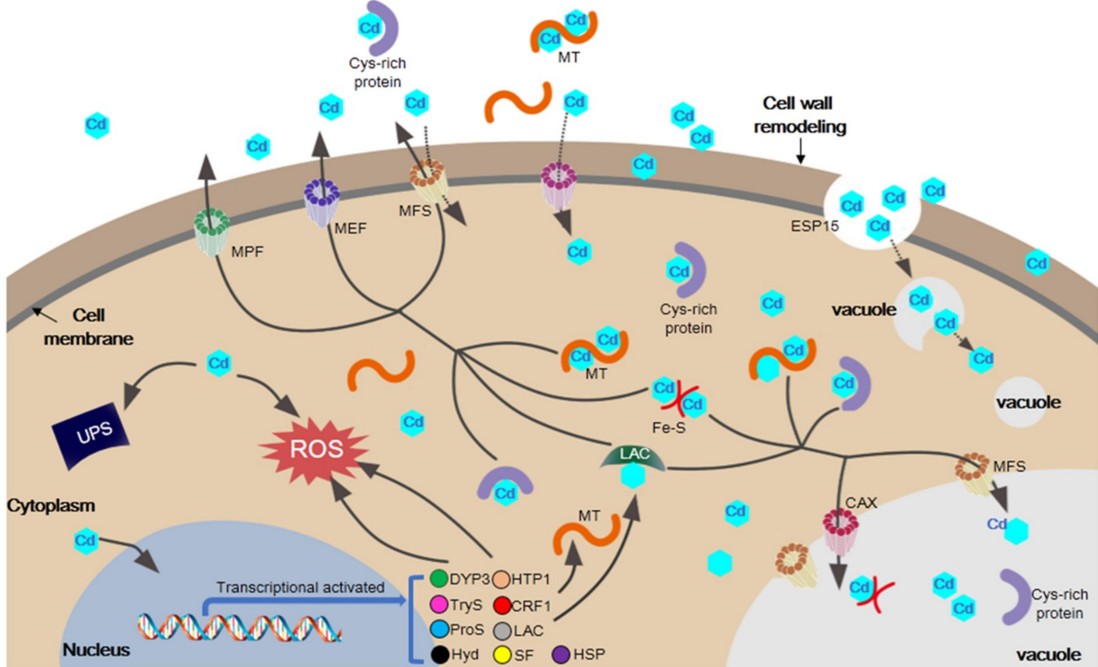

**Figure 6.** The schematic map of cellular response by *Pleurotus ostreatus* under Cd stress. Cd could be absorbed onto the surface of cell walls. Lots of Cd could be transported into cytoplasm by Transporters or Endocytosis (EPS15). Cysteine rich protein (Cyc-rich protein), Fe-S cluster (Fe-S), metallothionein (MT) and laccase (LAC) chelated Cd, and the metal chelation could be transported extracellular or into vacuole by membrane transporter metal-phosphate complexes efflux (MPE), Mate efflux family (MEF), major facilitator Superfamily (MFS) and $Ca^{(2+)}/H^{(+)}$ exchange (CAX). Reactive Oxygen Species (ROS) and ubiquitin-proteasome system (UPS) can be activated by heavy metals and some antioxidase such as DyP-type peroxidase (DYP3), heme-thiolate peroxidase (HTP1) could be induced. Copper resistance protein CRF1, a copper binding transcription factor, is more sensitive to Cd but not to other metals and would activate late metallothionein (MT) transcription directly. Macrofungi detoxify also by synthesis tryptophan (TryS) and proline (ProS), hydrophobin (Hyd), heat shock protein (HSP) and stress-inducible protein (SF).

　　Cell wall remolding was significantly induced under high Cd stress in the oyster mushroom fruiting body. A wide variety of cellular processes related to chitin modification, like carbohydrate esterase family 4, glycoside hydrolase family 18, glycosyltransferase family 4 [57–59], were upregulated in this study. Macrofungal cell wall, mainly composed of chitin [60], is known to participate in heavy metal detoxification [61]. Carboxyl, hydroxyl, amine, phosphate and sulfhydryl groups of chitin effectively sequestrate metals onto cell wall [62,63].

　　Some well-known metal stress genes were also found to be upregulated, like cysteine-rich related genes and laccase genes. Cd in microbial cells can bind to sulfide, cysteine, iron-sulfur centers (FeS centers) and thiol groups or replace other transition-metal cations from such sulfur-rich complex compounds [64,65]. Metallothionein (MT) is one of the most studied cysteine-rich proteins responsible

for Cd tolerance and accumulation in both prokaryotic and eukaryotic cells [66,67]. Upregulation of a copper resistance CRF1 homolog was detected in this study. CRF1 is sensitive to Cd but not to other metals and would activate MT transcription directly [68]. Tryptophan biosynthesis may also be involved in Cd detoxification in the oyster mushroom, since increase of tryptophan synthesis is linked to GSH release, which is a precursor for cysteine-rich metal chelators [64]. Meanwhile, laccase was able to chelate heavy metals and protect cell integrity. Laccase was found to be induced by Cd stress in previous studies in the ectomycorrhizal fungus *Paxillus involutus* [69].

**Supplementary Materials:** The following are available online at http://www.mdpi.com/2075-163X/8/12/589/s1, Figure S1. Total dry biomass of the fruiting body in the Cd treatments and control in this study (*n* = 3). Bars with different letters are significantly different at $p \leq 0.05$ (one-way ANOVA). Table S1: Quality control statistics of RNAseq by the software Sickle and SeqPrep. Table S2. The typical differentially expressed genes (DEGs) among the three treatments for heatmap. Table S3. COG classification of differentially expressed genes (DEGs) in the Cd treatments. Table S4. Typical upregulated genes of the 150 mg/kg Cd treatment.

**Author Contributions:** X.L. initiated the concept. X.L. and M.C. designed the mycoextraction system and the experiment. M.C. carried out the experiment. X.Z., S.Y. and L.C. helped in sampling and chemical analyses. L.W. and X.Z. helped in the transcriptomic analysis. J.H. provided help in metal analysis. M.C. and L.W. drafted the manuscript and all authors approved the final version.

**Funding:** This work was financially supported by the National Key Research and Development Program of China (2018YFD0800306), the National Natural Science Foundation of China (41877414), and Hebei Science Fund for Distinguished Young Scholars (D2018503005).

**Conflicts of Interest:** The authors declare no conflict of interest.

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
