# Peer review of "Mycoextraction: Rapid Cadmium Removal by Macrofungi-Based Technology from Alkaline Soil"

_minerals, doi:10.3390/min8120589_

Round 1
Reviewer 1 Report
Dear Authors,
You stated one technique of bioremediation using macrofungi to remove heavy metals from soils.
- Figure 1.
It is not designed professionally. Please, update its design and use relevant colors.
- Introduction
Please, add further information about Cd sources in the environment and effect in general. I can recommend the following references:
* Optimization of Cadmium (CD2+) removal from aqueous solutions by novel biosorbent
* Bio-based Methods for Wastewater Treatment: Green Sorbent
Materials and methods
- Line 103-104: Complete removal of the plastic bag coupled with underground plantation of the substrate with overgrown mycelia (i.e. mushroom bags) failed to develop fruiting body within a month under the standard conditions.
*Could you please state the reason for such failure? What was the main reason for the failure as stated in the previous literature?
- How could you prepare your samples to measure in ICP-MS? Please refer it to your manuscript.
Results
- Figure 2.
It is not clear what the figure legend refer to.
- Figure 3.
You state that the bars are from the ANNOVA analysis but it is always error bars. How did you apply ANNOVA test on such data?
Please, enhance the quality of this figure.
Discussion
- Did you design or draw figure 6 by yourself? if not, please, the reference.
-General remarks.
Could you please make a table to compare Cd removal with other techniques using fungi?
Best regards
Author Response
You stated one technique of bioremediation using macrofungi to remove heavy metals from soils.
- Figure 1.
It is not designed professionally. Please, update its design and use relevant colors.
The Authors: Thank you for this valuable suggestion. We have made substantial improvement on this figure.
- Introduction
Please, add further information about Cd sources in the environment and effect in general. I can recommend the following references:
* Optimization of Cadmium (CD2+) removal from aqueous solutions by novel biosorbent
* Bio-based Methods for Wastewater Treatment: Green Sorbent
The Authors: Thank you for this suggestion. We have added more descriptions (Line 48-53 ) on Cd pollution citing the recommended references.
Materials and methods
- Line 103-104: Complete removal of the plastic bag coupled with underground plantation of the substrate with overgrown mycelia (i.e. mushroom bags) failed to develop fruiting body within a month under the standard conditions.
*Could you please state the reason for such failure? What was the main reason for the failure as stated in the previous literature?
The Authors: Underground cultivation of mushroom always requires a long time for fruiting body formation. Meanwhile, fruiting body formation requires stringent environmental stimulations for underground cultivation. This has been observed in at least two previous studies.
- How could you prepare your samples to measure in ICP-MS? Please refer it to your manuscript.
The Authors: Thank you for this valuable suggestion. We have provided addition description (Line 164-167) in the manuscript.
Results
- Figure 2.
It is not clear what the figure legend refer to.
The Authors: We have added more to explain the figure. Aging curve is a standard expression in ecotoxicology, to describe the decline of metal bioavailability in soil.
- Figure 3.
You state that the bars are from the ANNOVA analysis but it is always error bars. How did you apply ANNOVA test on such data?
The Authors: We did ANNOVA analysis using SPSS. Yet we are not clear what is required here, as the superscripts have been added to indicate the results of ANNOVA test.
Please, enhance the quality of this figure.
The Authors: Thanks for this comment. We have figures of 600 dpi if the editor requires them.
Discussion
- Did you design or draw figure 6 by yourself? if not, please, the reference.
The Authors: All the figures are based on the results of this study. Thanks.
-General remarks.
Could you please make a table to compare Cd removal with other techniques using fungi?
The Authors: In Table 2 we have compared several studies using plants with our fungi-based system. Hopefully this can provide useful information to the readers.
Best regards

Reviewer 2 Report
I recommend publication of manuscript entitled „ Mycoextraction: Rapid Cadmium Removal by Macrofungi-Based Technology from Alkaline Soil” in Minerals.
I found the paper very interesting. Fungi are a promising alternative for bioremediation of
heavy metal-contaminated environments and can also be successfully applied for mycoextraction of cadmium ions form soil. In the manuscript, very interesting, innovatory macrofungi‐based system was designed for rapid cadmium removal from alkaline soil. For the studies highly Cd tolerant Pleurotus ostreatus strain was applied. Authors applied several techniques for detailed characteristic of soil and cadmium stability, which are a very crucial stages of the work. Besides investigation of cadmium removal efficiency, detailed gene characteristic was done. The results of the studies are very promising due to obtained efficiency of annual cadmium removal.
The manuscript is generally well written. Experiments are well planed and very extensive. Manuscript includes many valuable data and interesting discussion. The great number of varied data, detailed characteristic of samples, comprehensive discussion of results and indication of direction of further studies improve the quality of the work.
I have not found anything that would require any revision.
Author Response
The Authors: Thank you very much for reviewing our manuscript and considering it “very interesting”.
Reviewer 3 Report
Authors have to revise article:
1- Please improve kinetics studies of Cd uptake
2- Authors can depict Cd speciation
3- Authors can build up Equilibrium isotherms related to this particular system
Author Response
Authors have to revise article:
1- Please improve kinetics studies of Cd uptake
2- Authors can depict Cd speciation
3- Authors can build up Equilibrium isotherms related to this particular system
The Authors: Thank you very much for reviewing our manuscript. And we appreciate that you point out very important research directions to improve our system. As you may see, fungi are promising materials for metal biosorption, but mycoextraction is really new and proposed firstly in our study. We are happy to carry out further studies, including those proposed here by you. And sincerely hope you can review our future reports on mycoextraction.
Round 2
Reviewer 3 Report
Paper has been improved and can be accepted.